# Measurement of Dynamic Elastic Modulus and Poisson’s Ratio of Chemically Strengthened Glass

**DOI:** 10.3390/ma13245644

**Published:** 2020-12-10

**Authors:** Sun-Youn Ryou, Chang-Soon Lee, In-Sik Cho, Auezhan Amanov

**Affiliations:** 1Department of Advanced Materials Engineering, Sun Moon University, Asan 31460, Korea; sunglass@sunmoon.ac.kr (S.-Y.R.); lcs8769@sunmoon.ac.kr (C.-S.L.); mbrosia1018@naver.com (I.-S.C.); 2Department of Mechanical Engineering, Sun Moon University, Asan 31460, Korea; 3Fusion Science and Technology, Sun Moon University, Asan 31460, Korea

**Keywords:** ultrasonic pulse echo overlap technique, impulse excitation technique, dynamic elastic modulus, glass, ion exchange

## Abstract

Glass with strong durability and transparency has been in the spotlight in various fields, including displays. Elastic and shear moduli and Poisson’s ratio are important properties of glasses. The purpose of this study is to evaluate the change in mechanical properties, such as the dynamic elastic modulus and Poisson’s ratio, with respect to the chemical strengthening time of glass for display applications, as measured by static and dynamic methods. The basic measurement principle of the dynamic method is to measure acoustic speed or resonant frequency using an ultrasonic generator. The mechanical properties of both non-strengthened and chemically strengthened glasses were investigated. It was found that the strength of the chemically strengthened glass decreased when chemical strengthening time increased. Chemical strengthening increased the bending strength and decreased the elastic modulus due to the introduction of compressive residual stress at the surface.

## 1. Introduction

Glass is widely used not only for aircraft screens, military vehicles, cars and high-speed trains but also for the design of earthquake-resistant buildings exposed to harsh external environments. This is because of its high impact resistance and excellent optical properties [1,2]. In addition, nowadays, bio-glasses are also being used in medicine, particularly for bone tissue engineering [3,4]. An increase in demand for the usage of various applications has led to thin glass with high strength and high scratch-resistance being developed through a variety of physical and chemical strengthening methods [5,6]. Mechanical properties such as the dynamic elastic modulus and Poisson’s ratio are essential data points to consider for the improvement of glass quality. In the field of fingerprint recognition, the acoustic speed data obtained from ultrasonic sensors are also important. In addition, the strength of glass may be increased intentionally by inducing compressive stresses to the surface. A physical glass strengthening method can induce a stress distribution by using differences in the cooling speed between the surface and interior through thermal tempering [7]. Chemical strengthening replaces small Na^+^ ions with large K^+^ ions on the surface of the glass through the temperature and time control of molten potassium nitrate, which results in the formation of compressive stress layers [8,9,10]. Various mechanical testing methods, such as three-point bending and four-point bending tests, are usually used for evaluating the mechanical properties in bending, while dynamic failure subjected to vibrational compressive stress can be evaluated by dynamic test methods [10]. Moreover, besides these testing methods, a computer simulation method can be also used to investigate the failure and properties of strengthened glasses at the atomic level [11,12,13,14].

The elastic modulus of glasses can be measured by using the slope and maximum stress values of the load versus displacement graph obtained by a bending test [15,16]. Furthermore, the mechanical properties of strengthened glass can be also measured by using a flexural dynamic bending test [17]. It is worth mentioning here that there are two different methods of bending test. These are ‘static’, which is an application of mechanical load, and ‘dynamic’, which uses an ultrasonic sound velocity or resonant frequency for measuring the elastic modulus of glasses. The latter test has the advantage of experiencing fewer measurement errors and a shorter testing time compared to the former [18]. In this paper, the mechanical properties such as elastic modulus and Poisson’s ratio of the non-strengthened glass were evaluated using the traditional static three-point bending test (BT); the ultrasonic pulse echo overlap technique (UPET); and the impulse excitation technique (IET) dynamic test methods, while the mechanical properties of the chemically strengthened glass were evaluated using an IET dynamic test method.

## 2. Experimental Investigations

### 2.1. Three Point Bending Method

In this study, a Corning^®^ Gorilla glass 3 with dimensions of 75 × 50 × 0.7 mm^3^ was used as a specimen. Ion exchange treatment was performed in a molten potassium nitrate (KNO_3_) bath held at 420 °C for varying periods of 1, 2 and 3 h. All the specimens in each temperature condition were prepared in separate batches. After the ion exchange, the glasses were removed and cooled, and then rinsed with deionized (DI) water. The three-point bending test was performed according to the ASTM C158 test method [15], which tests flexure strength and is designed to minimize breaks that initiate at the edge. The test specimens were plate-like, rather than beam-like, but a beam stress theory is commonly used for calculating strength from an experimentally obtained maximum load in the three-point bending test [19]. In this paper, the mechanical properties of thin glass specimens closely resembling the actual product are correlated based on the ASTM C158 and Equation (2) below. This allowed us to evaluate the basic relational analysis of elastic modulus and Poisson’s ratio. The test standards were established in a precise order, considering the closest test method, bending speed, etc. As shown in Figure 1, a three-point bending test was conducted by supporting the glass at points A and B and applying a load at the center. The test conditions were set at a control speed of 0.1 mm/s and a distance of 30 mm between the supporting points to ensure a stable slope as much as possible. The edge stress of the test glass specimen was simulated by finite element analysis (FEA) during the three-point bending test to assure if failure initiated at the edge. FEA results (not shown here) provided local stress failure results. It was also confirmed that the failure location was found to be an edge, where the failure actually occurred at the point of maximum stress. Moreover, local stress failures can be calculated by Griffith theory, which considers the elastic modulus, specific surface energy and one-half of the internal crack length [20].

### 2.2. Elastic and Shear Moduli of Bending Method

The elastic modulus by bending test can be obtained from slope and length of the specimen as shown in Equation (1) by using the cross-sectional second moment as expressed in Equation (2) in case of square specimen [21]:(1)$$E\;=\;\left(\frac{F}{\Delta }\right)\frac{L^{3}}{48I}$$
(2)$$I\;=\;\frac{bh^{3}}{12}$$
where *E* is the Young’s modulus; $\left(\frac{F}{∆}\right)$ is the slope of the tangent to the initial straight-line portion of the load-deflection curve; *L* is the length of support; *I* is the moment of inertia; *b* is the width of specimen; and *h* is the thickness of the specimen.

Shear modulus can be obtained using a slope of displacement–load graph, the length of specimen, and the cross-sectional area from the following Equations (3) and (4), similar to the elastic modulus:(3)$$G\;=\;\left(\frac{F}{∆}\right)\frac{L^{3}}{4A}$$
(4)$$\nu \;=\;\left(\frac{E}{2G}\right)-1$$
where *G* is the shear modulus, *A* is the cross-section area and *ν* is the Poisson’s ratio and it can be obtained from Equation (4) using elastic and shear moduli obtained from Equations (1) and (3), respectively.

### 2.3. Ultrasonic Pulse Echo Overlap Technique

Figure 2 shows the basic measurement system and device for UPET technique. Ultrasonic echo sound velocities *V_l_* and *V_t_* are measured using two ultrasonic transducers, respectively. *V_l_* is the longitudinal wave velocity, *V_t_* is the transverse wave velocity, and ultrasonic wave velocity is calculated from the round trip distance divided by the round trip time (*t_l_* and *t_t_*) of ultrasonic waves generated by the ultrasonic transducer. The relationship expressions for obtaining elastic and shear moduli, and Poisson’s ratio are as follows:(5)$$\nu_{l}\;=\;\frac{2h}{t_{l}}$$
(6)$$\nu_{l}\;=\;\frac{2h}{t_{t}}$$
where *h* is the thickness of the specimen being measured.

The velocities obtained by Equations (5) and (6) may be introduced in Equation (7) to obtain Poisson’s ratio:(7)$$V\;=\;\frac{1-2\left(\nu_{t}^{2}-\nu_{l}^{2}\right)}{2-2\left(\nu_{t}^{2}-\nu_{l}^{2}\right)}$$

The shear modulus *G* is obtained by substituting the lateral wave velocity *V_t_* and the density *ρ* of the test material into Equation (8), and the elastic modulus *E* can be arranged according to Equation (9):(8)$$G=\rho V_{t}^{2}$$
(9)E=2G(1+ν)

### 2.4. Impulse Excitation Technique

The impulse excitation technique (IET) is designed to find the resonant frequency and to measure the elastic modulus of the material by striking through an impulse tool. The flexural frequency is measured by placing the specimen parallel to the node line and striking the specimen using the impulse tool on the impulse point as shown in Figure 3a, and analyzing the signal through the microphone. Similarly, the shear modulus is determined by measuring the flexural frequency and torsional frequency by placing the node line cross–wise, as shown in Figure 3b.

In addition, the thickness, length, width, and weight of the test specimens can be substituted in Equations (10) and (11) to obtain the dynamic elastic and shear moduli [22]:(10)$$E=0.9465\left(\frac{mf_{f}{}^{2}}{b}\right)\left(\frac{L^{3}}{t^{3}}\right)T$$
(11)$$G\;=\;\frac{4Lmf_{t}{}^{2}}{bt}\left[\frac{B}{1+A}\right]$$
where *m* is the weight of the specimen; *f_f_* is the flexure frequency; *f_t_* is the torsional frequency; *b* is the width of the specimen; *L* is the length of the specimen; *t* is the thickness of the specimen; and *T*, *B* and *A* are correction factors. *T*, *B*, and *A* are the correction factor values related to the dimensions of the specimen in both calculation and analysis equations. The correction factors are well described in ASTM E1876 standard [22]. About ten specimens were prepared for each measurement and all the experimental tests were repeated at least three times to get reliable data within standard deviation.

## 3. Results and Discussions

### 3.1. Bending Test

Figure 4 shows the bending stress vs. displacement curve of the non-strengthened glass. It is clear from the graph that the non-strengthened glass showed a maximum breaking strength of 125 MPa and a displacement of 0.87 mm. The slope measured with F/Δ was found to be 22.67 ± 3.45. By substituting into Equations (3) and (4), the elastic and shear moduli, and Poisson’s ratio were found to be 71.40 ± 11.34 GPa, 27.00 ± 4.87 GPa and 0.22 ± 0.036, as listed in Table 1, respectively. This result is in agreement with the result previously reported in the literature [23]. In addition, it is important to mention here that the measurements were repeated at least 3–4 times in order to obtain reliable data within standard deviation.

### 3.2. UPET and IET Tests

Mechanical properties of the non-strengthened glass were measured by UPET and IET methods. Figure 5 shows the results of the speed measurements of each longitudinal and transverse wave obtained by UPET. The measurement results revealed that the speed of longitudinal and transverse waves were found to be 0.97 ± 0.015 and 1.58 ± 0.032 μs, respectively. The Poisson’s ratio, dynamic elastic and shear moduli were calculated by substituting into Equations (7)–(9), as 0.197 ± 0.015, 72.41 ± 11.67 GPa and 30.23 ± 4.68 GPa, as outlined in Table 2, respectively. Figure 6 shows the flexural and torsional frequencies through FFT analysis according to the time series analysis measured using the IET method. The flexural and torsional frequencies were found to be 1386 ± 270 and 1862 ± 340 Hz, respectively. Based on these calculations, the dynamic elastic and shear moduli were verified as 72.52 ± 11.71 and 30.43 ± 4.76 GPa by substituting them into Equations (10) and (11), respectively, as listed in Table 3. The measurements were repeated in order to obtain reliable data within standard deviation as listed in Table 2 and Table 3.

Table 4 summarizes the results of elastic and shear moduli measured by using dynamic UPET and IET methods. The elastic and shear moduli obtained by UPET and IET methods were found to be approximately 71.40 ± 11.34 and 30.23 ± 4.68, 72.52 ± 11.71 and 30.43 ± 4.76 GPa, respectively, which are higher compared to the elastic and shear moduli obtained by static bending test method. In addition, the Poisson’s ratio was found to be 0.22 ± 0.036 for the bending and approximately 0.197 ± 0.015 and 0.191 ± 0.024 for the UPET and IET test methods, respectively. This discrepancy may be explained by the fact that the flexure and shear stress modes for static bending tests are measured simultaneously, while dynamic acoustic and resonance measurements differ in theoretical content when compared to an atomic-combined energy or spring model [14]. It should be mentioned here that although the results of the static method and the dynamic method do not differ greatly, the results of the dynamic acoustic method showed a slightly larger error range than the static mechanical method. This can be explained by the difference in the computational interpretation used for the physical measurement method. In addition, the BT method calculates and analyses data according to the bending intensity gradient. In contrast, the UPET method is different from the longitudinal wave and transverse wave sensors in that the ultrasonic sensors represent the difference in time or velocity of the reverberations. Finally, the IET method considers the natural frequency value of acoustic transmission between the longitudinal wave and the transverse wave. By deriving and analyzing the calculation, the static method can be described as calculating the bonding energy between atoms, and the dynamic method can be described as the spring model between atoms.

### 3.3. Chemically Strengthened Glass Results

The strengthening time was 1, 2 and 3 h, with all glass being strengthened under the same temperature. The glass was then cooled, which resulted in the surface of the glass becoming compressed, while the center core remained in a state of tension. The strength of the glass increased after chemical strengthening compared to the non-strengthened glass, but it was decreased by increasing the chemical strengthening times of 1, 2 and 3 h from 668.3 ± 88.43, 645.5 ± 82.37 and 623.9 ± 80.49 MPa, respectively, as shown in Figure 7. Significantly, the bending strength was observed to form compressive stresses at the surface when the chemical strengthening mechanism of a Na^+^ ion was substituted with a K^+^ ion. However, the longer the chemical strengthening time conditions, the more likely it was that the bending strength decreased by about 20 MPa (see Table 5). These results are related to the loss of surface compressive stress, which may be attributed to the difference in temperature when cooled to ambient temperature during ion exchange [19,24,25,26]. In general, high compressive stress at the surface and low tension at the center can improve the strength of chemically strengthened glass. Moreover, Kerper and Scuderi investigated the mechanical properties of chemically strengthened glasses at high temperatures [27]. It was found that the elastic modulus decreased with increasing temperature, with a sharp inflection slightly above room temperature. The ion exchange of K^+^ ions by chemical strengthening layer is about 200 µm. In the BT method, the mechanical strength value is the compressive stress of the surface layer, resulting in a difference value depending on the chemical strengthening treatment conditions. In the UPET method, there was a limit in measuring the difference in acoustic speed for both sides of the 200 µm, so there was some difference in the error range, as in the BT method. In other words, it was confirmed that both the BT and UPET measurements allowed for the derivation of measurement errors. Moreover, the IET method was able to measure minute variations in natural frequency of the glass.

Table 6 compares the dynamic elastic modulus values using an IET method in the same conditions of chemical strengthening, showing that it was about 3 to 5 GPa lower than that of the non-strengthened glass. In addition, the longer the strengthening time, the lower the elastic modulus value. This is the physical meaning of elastic modulus, and it is known that Equation (12) is formed in relation to the atomic force [28]:(12)$$F\;=\;-\frac{dV\left(∆r\right)}{d\left(∆r\right)}\;=\;-{\left(\frac{d^{2}V}{dr^{2}}\right)}_{r_{0}}∆r$$
where: *F* is the atomic force; Δ*r* (*r* − *r*_0_) is the movement between atoms; *r*_0_ is the equilibrium interatomic spacing.

In other words, based on the spring model according to the ion exchange of K^+^ ions by chemical strengthening, the difference in elastic waves can be seen by the interaction of atomic forces as shown in Figure 7, which indicates that the K^+^ (0.231 nm, 63 °C) ions have a larger atomic radius and a lower melting point than Na^+^ (0.186 nm, 98 °C). It was previously reported that the exchange of K^+^ ions by strengthening has two competing processes: one is the generation of stress from “stuffing” large foreign ions into small host ion sites in the glass, and the other is the relaxation of stress using viscous flow [29]. The exchange of large ions such as K^+^ for comparatively smaller ions such as Na^+^ in glass at temperatures below the material’s strain point leaves the surface of the glass in a state of high compression. Because glass products usually break due to excessively applied tension acting on a surface flaw, the introduction of a high surface compression strengthens the glass [30]. Furthermore, the chemically strengthened glass has a compressive stress layer formed on the surface by ion exchange treatment (see Figure 8a). Distribution of K^+^ and Na^+^ ions after heat treatment at 420 °C for 3 h is shown in Figure 8b,c. It is obvious that the flow depth of K^+^ and Na^+^ ions was less than 10 µm. The surface of the glass is ion–exchanged to form a surface layer in which compressive stress remains. Specifically, a metal ion with a small ionic radius which exists near the glass plate’s surface is converted to an ion with a larger ionic radius by ion exchange at a temperature lower than the glass transition point, as illustrated in Figure 9. The exchange depth, the concentration of ions that can be replaced with large ions, and any stress or structural relaxation that may occur determine the magnitudes of the residual stress. Thereby, compressive stress remains on the surface of the glass, meaning the strength of the glass can be increased. Unlike tempered glass, the surface layer is thin with a high stress at the surface. The tensile stress inside the glass is low and almost constant with thickness. The maximum compressive stress at the surface can be calculated by Equation (13) below:(13)$$\sigma_{max}\;=\;\frac{E}{1-\upsilon }\frac{\Delta V}{3V}$$
where *E* is the elastic modulus; *ν* is the Poisson’s ratio; Δ*V*/*V* is the relative volume increase due to an exchange of smaller ions for ions with larger sizes. Hence, understanding the effect of compressive residual stress on glass fractures is not only of long-standing fundamental interest, but also important for controlling the mechanical properties of glass products.

## 4. Conclusions

In this paper, mechanical properties such as the dynamic elastic modulus and Poisson’s ratio, with respect to chemical strengthening time of glass, were measured by static and dynamic methods. The static bending test confirmed that the bending strength of the non-strengthened glass used in this study was 138.0 ± 14.64 MPa, while the elastic and shear moduli were 71.40 ± 11.34 and 27.00 ± 4.87 GPa, respectively. Comparing the bending strengths of the non-strengthened and chemically strengthened glasses with KNO_3_—1 h of glass showed that the bending strength was increased by more than four times from 138.0 ± 14.64 to 668.3 ± 88.43 MPa. In addition, the bending strength tended to decrease when increasing the chemical strengthening time. Chemical strengthening increased the bending strength and decreased the elastic modulus due to the introduction of compressive residual stress at the surface.

## Figures and Tables

**Figure 1 materials-13-05644-f001:**
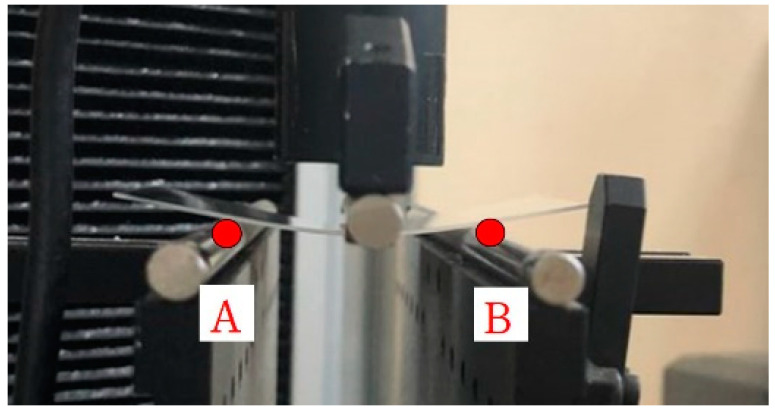
Three point bending test machine.

**Figure 2 materials-13-05644-f002:**
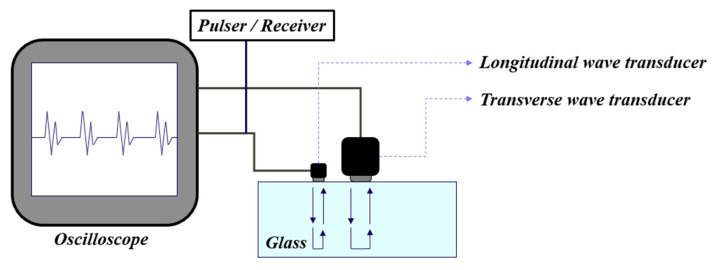
Schematic representation of ultrasonic pulse echo overlap technique.

**Figure 3 materials-13-05644-f003:**
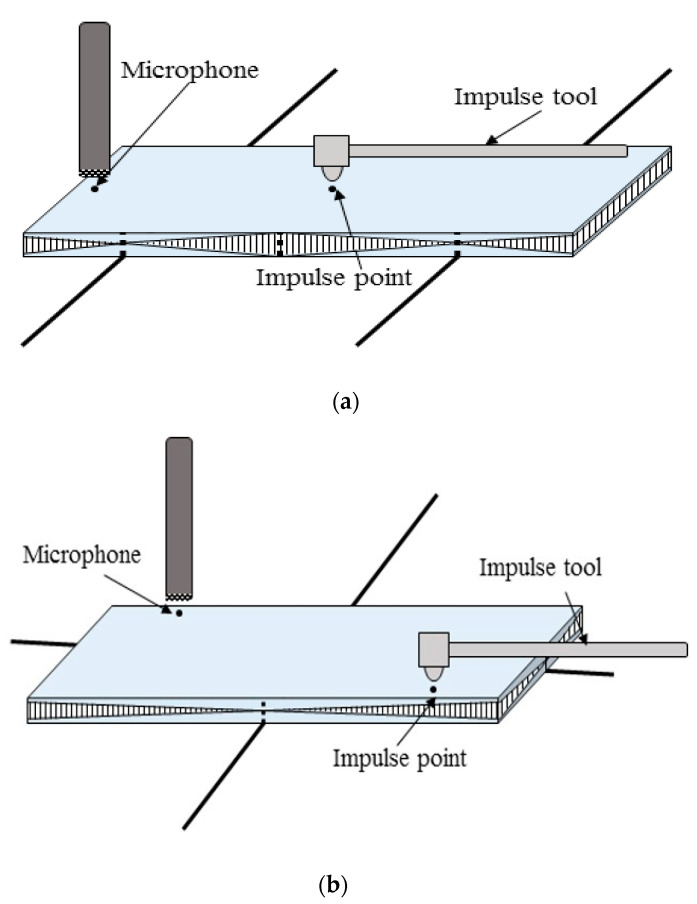
Flexural test in IET (**a**) and torsional test in IET (**b**).

**Figure 4 materials-13-05644-f004:**
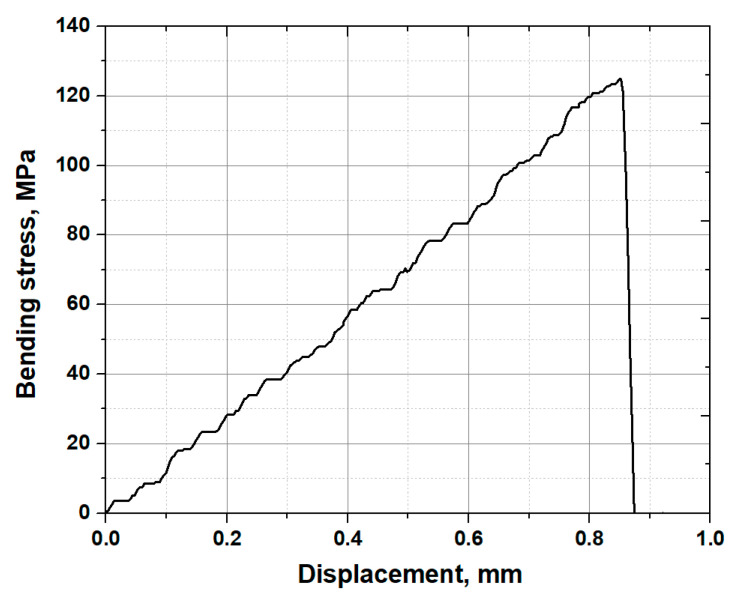
Bending test result of the non-strengthened glass.

**Figure 5 materials-13-05644-f005:**
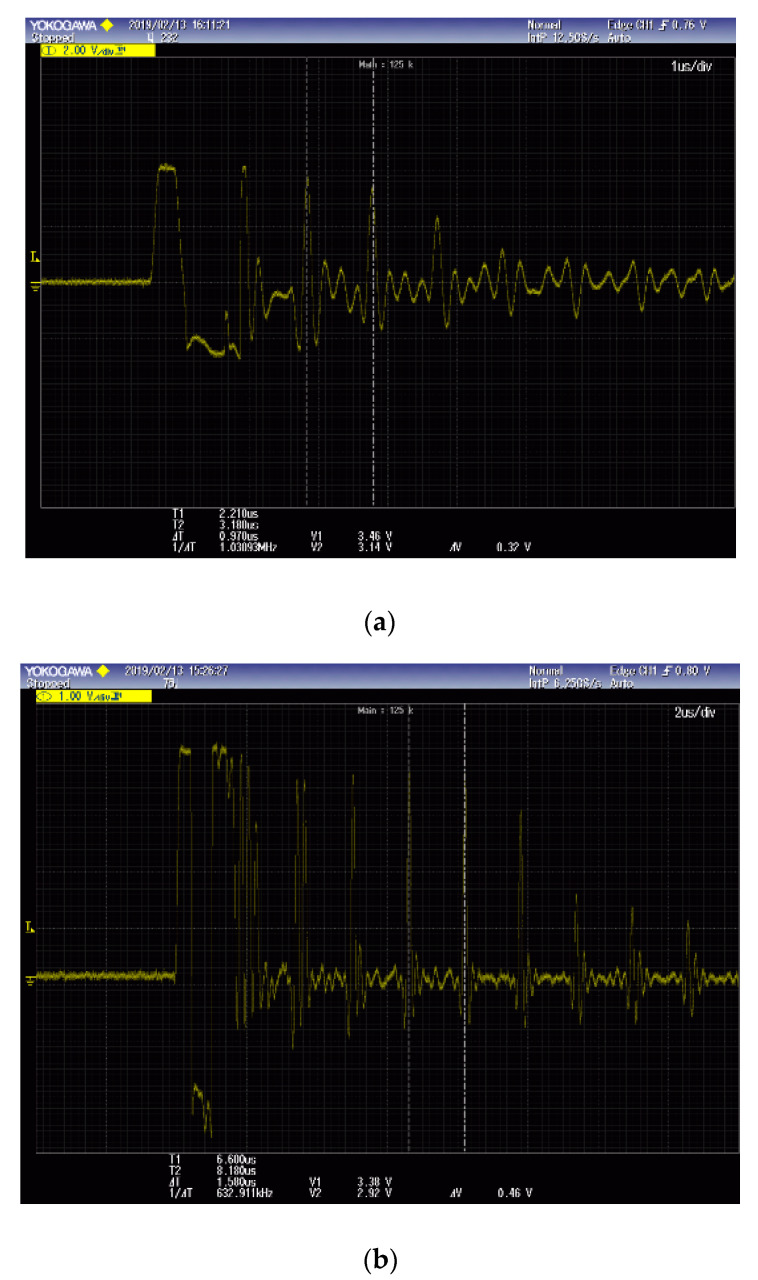
Ultrasonic echo pulse measurements: longitudinal wave (**a**), transverse wave (**b**).

**Figure 6 materials-13-05644-f006:**
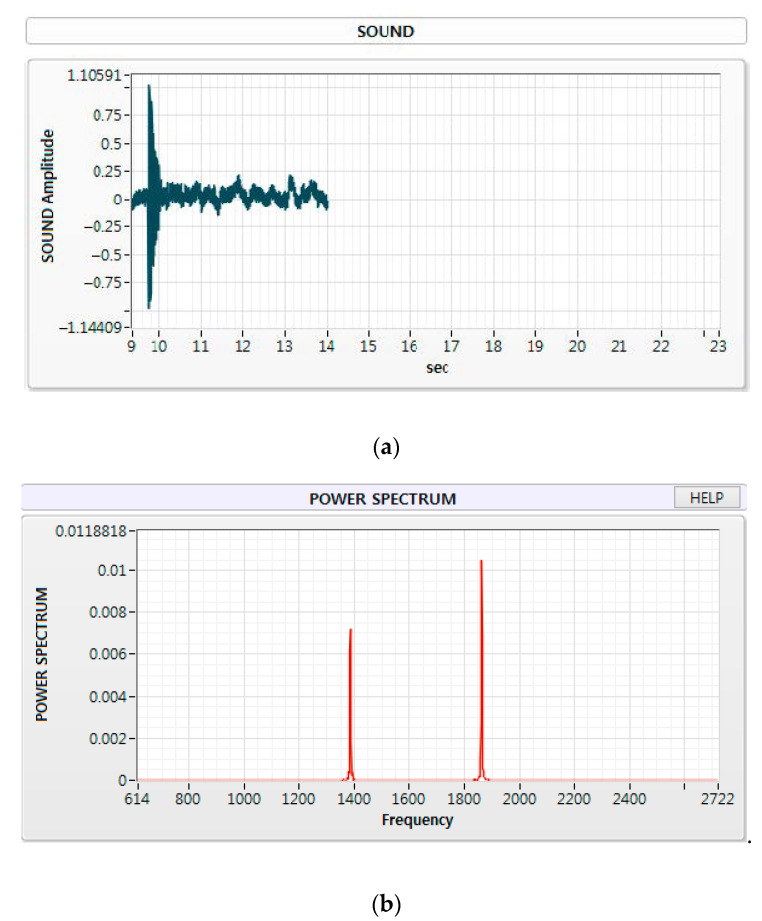
IET time series analysis (**a**) and power spectrum analysis (**b**) results.

**Figure 7 materials-13-05644-f007:**
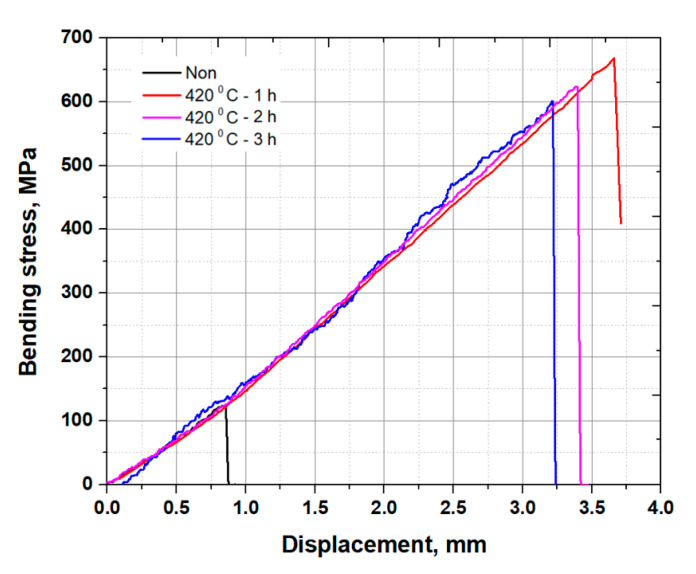
Bending test results of the non-strengthened and chemically strengthened glasses.

**Figure 8 materials-13-05644-f008:**
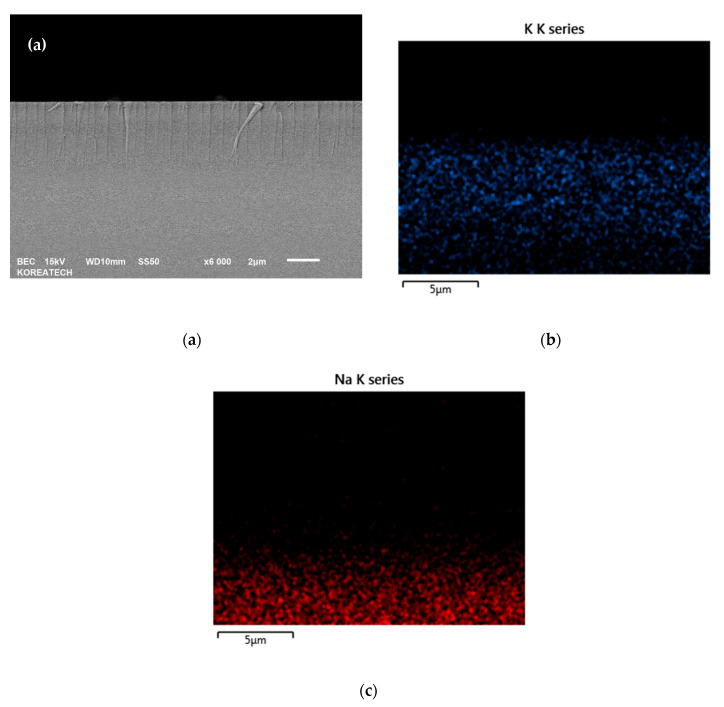
Cross-sectional SEM (**a**) and distribution of K^+^ (**b**) and Na^+^ (**c**) ions images of the glass heat treated at 420 °C for 3 h.

**Figure 9 materials-13-05644-f009:**
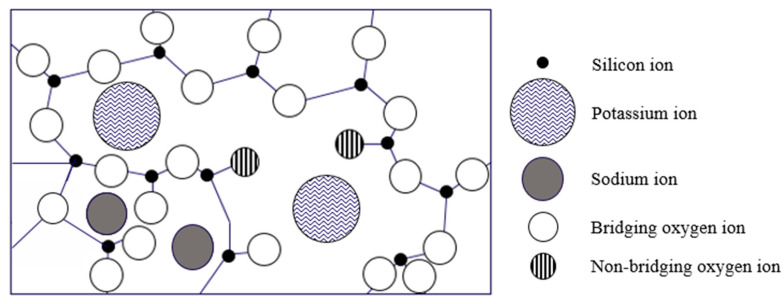
Schematic diagram of an ion exchange.

**Table 1 materials-13-05644-t001:** Test results of *E*, *G* and *ν* obtained via bending test.

*F*/Δ	*E*, GPa	*G*, GPa	*ν*	Bending Stress, MPa
22.69 ± 3.45	71.42 ± 11.34	27.12 ± 4.87	0.22 ± 0.1	138.1 ± 14.64

**Table 2 materials-13-05644-t002:** Test results of *E*, *G* and *ν* in UPET.

*V_ι_*, µs	*V_t_*, µs	*E*, GPa	*G*, GPa	*ν*
0.97 ± 0.015	1.58 ± 0.032	72.41 ± 11.67	30.23 ± 4.68	0.197 ± 0.028

**Table 3 materials-13-05644-t003:** Test results of *E*, *G* and *ν* in IET.

Flexural (*F*_1_)*,* Hz	Torsional (*F*_2_)*,* Hz	*E,* GPa	*G,* GPa	*ν*
1386 ± 270	1862 ± 340	72.52 ± 11.71	30.43 ± 4.76	0.191 ± 0.024

**Table 4 materials-13-05644-t004:** Test results of *E*, *G* and *ν*.

Methods	BT	UPET	IET
*E*, GPa	71.40 ± 11.34	72.41 ± 11.67	72.52 ± 11.71
*G*, GPa	27.00 ± 4.87	30.23 ± 4.68	30.43 ± 4.76
*ν*	0.22 ± 0.036	0.197 ± 0.028	0.191 ± 0.024

**Table 5 materials-13-05644-t005:** Results obtained by bending test.

Chemical Strengthening Conditions		Bending Strength, MPa
Non	Non	138.0 ± 14.64
KNO_3_	420 °C-1 h	668.3 ± 88.43
	420 °C-2 h	645.5 ± 82.37
	420 °C-3 h	623.9 ± 80.49

**Table 6 materials-13-05644-t006:** Results obtained by IET test.

Glasses	Flexural (*F*_1_), Hz	Torsional (*F*_2_), Hz	*E*, GPa	*G*, GPa	*ν*
Non	1386 ± 270	1862 ± 340	72.52 ± 11.71	30.43 ± 4.76	0.19 ± 0.024
KNO_3_ 420 °C-1 h	694 ± 89.41	665 ± 86.11	69.25 ± 10.48	30.73 ± 4.89	0.13 ± 0.019
KNO_3_ 420 °C-2 h	693 ± 89.37	665 ± 86.15	68.16 ± 10.35	30.45 ± 4.74	0.12 ± 0.018
KNO_3_ 420 °C-3 h	690 ± 89.03	664 ± 86.08	67.69 ± 10.31	30.31 ± 4.69	0.12 ± 0.018

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
