# Peer review of "Measurement of Dynamic Elastic Modulus and Poisson’s Ratio of Chemically Strengthened Glass"

_materials, 2020, doi:10.3390/ma13245644_

Round 1

Reviewer 1 Report

The manuscript related to study of elastic modulus and poisson ratio of chemically strengthened glass, is well written and interesting for readers. The introduction is well written, and results are well explained. 

I would like to recommend this article for publication. A small change can be beneficial in my opinion before final publication. 

- Authors have written different equation at same line, its better to write one equation per line. 

- It is difficult to read figure 5, kindly used better image quality or enlarge size. 

Author Response

Comments and Suggestions for Authors

The manuscript related to study of elastic modulus and poisson ratio of chemically strengthened glass, is well written and interesting for readers. The introduction is well written, and results are well explained. 

I would like to recommend this article for publication. A small change can be beneficial in my opinion before final publication. 

- Authors have written different equation at same line, its better to write one equation per line. 

RESPONSE: One equation per line has been written in the revised manuscript.

- It is difficult to read figure 5, kindly used better image quality or enlarge size. 

RESPONSE: Fig. 5 has been enlarged in the revised manuscript to improve the readability.

Reviewer 2 Report

In this manuscript, Amanov and coworkers described study on measurement of dynamic elastic modulus and Poisson`s ratio of chemically strengthened glass. The static bending test confirmed that the bending strength of the non-strengthened glass used in that study was 138 MPa, while the elastic and shear moduli were 71 GPa and 27 GPa, respectively. Comparing the bending strengths of the non-strengthened and chemically strengthened glasses with KNO3 of glass showed that the bending strength was increased by more than 4 times from 138 MPa to 668 MPa. Also the bending strength tended to decrease with increasing the chemical strengthening time. Chemical strengthening increased the bending strength and decreased the elastic modulus due to the introduction of compressive residual stress at the surface.

Further comments:

-In the introduction, the authors mention the use of glass as: “Glass is widely used not only for aircraft screens, military vehicles, cars and high-speed trains but also for design of earth quake-resistant building exposed to harsh external environments because of high impact resistance and excellent optical property”. This is a very basic application, this sentence does not provide the reader with any new knowledge. What, for example, about bioglass, their use in medicine? for example: materials science and engineering: C 2019, 94, 516–523; micron 2018, 119, 64–71.

-Figure 5 is bad quality. It cannot be read.

-To indicate a positive ion (cation) like Na+, K+: Enter the plus sign (+) in superscripts, for KNO3, 3 should be subscript.

- Uncertainties are given with wrong decimal places. For example: page 7, line 218: 668.3±88.43, 645.5±82.37 and 623.9±80.49 MPa.

- It would also be interesting to study the chemical composition and morphology of the glass after ion exchange KNO3, e.g. using SEM microscopes with the EDS technique. If this technique is not available then ICP can be used, at present it is not known how many ions have been replaced.

-The structure of the glass can be investigated by IR, analyzing the vibration of the SiO bonds.

Author Response

Reviewer 2:

Comments and Suggestions for Authors

In this manuscript, Amanov and coworkers described study on measurement of dynamic elastic modulus and Poisson`s ratio of chemically strengthened glass. The static bending test confirmed that the bending strength of the non-strengthened glass used in that study was 138 MPa, while the elastic and shear moduli were 71 GPa and 27 GPa, respectively. Comparing the bending strengths of the non-strengthened and chemically strengthened glasses with KNO3 of glass showed that the bending strength was increased by more than 4 times from 138 MPa to 668 MPa. Also the bending strength tended to decrease with increasing the chemical strengthening time. Chemical strengthening increased the bending strength and decreased the elastic modulus due to the introduction of compressive residual stress at the surface.

Further comments:

-In the introduction, the authors mention the use of glass as: “Glass is widely used not only for aircraft screens, military vehicles, cars and high-speed trains but also for design of earth quake-resistant building exposed to harsh external environments because of high impact resistance and excellent optical property”. This is a very basic application, this sentence does not provide the reader with any new knowledge. What, for example, about bioglass, their use in medicine? for example: materials science and engineering: C 2019, 94, 516–523; micron 2018, 119, 64–71.

RESPONSE: Bioglass application in medicine has been added to the revised manuscript citing a couple of reference papers suggested by the reviewer.  

- Figure 5 is bad quality. It cannot be read.

RESPONSE: Fig. 5 has been enlarged in the revised manuscript to improve the readability.

-To indicate a positive ion (cation) like Na+, K+: Enter the plus sign (+) in superscripts, for KNO3, 3 should be subscript.

RESPONSE: Thank you for your comment. The correction has been made in the revised manuscript.

- Uncertainties are given with wrong decimal places. For example: page 7, line 218: 668.3±88.43, 645.5±82.37 and 623.9±80.49 MPa.

RESPONSE: The correction in Fig. 7 has been made in the revised manuscript.

- It would also be interesting to study the chemical composition and morphology of the glass after ion exchange KNO3, e.g. using SEM microscopes with the EDS technique. If this technique is not available then ICP can be used, at present it is not known how many ions have been replaced.

RESPONSE: The chemical composition and morphology of the glass after ion exchange KNO3 have been investigated by SEM+EDX and the result of KNO3 @ 420 C – 1 h has been added in the revised manuscript.

-The structure of the glass can be investigated by IR, analyzing the vibration of the SiO bonds.

RESPONSE: Thank you for your kind suggestion. We will consider it in our future investigations with the chemically strengthened glass.

Reviewer 3 Report

Your article entitled "Measurement of Dynamic Elastic Modulus and 2 Poisson`s Ratio of Chemically Strengthened Glass" is interesting to be published in Materials as the stretching and bending properties of chemically enhanced glass can be used for many applications. However, I found that your manuscript needs to be revised with minor points so that it can be published in this journal.

  1. Why in the beginning you explained three methods (BT, UPET, IET) to determine E, G, and v while the chemically strengthened glass results were only determined by IET. This should be clear in the introduction and Page 7, line 213.
  2. 2. Consistent with decimal points behind the comma. As we see in Table 1 (v and bending stress), it is not clear whether it is 1, 2, or 3 behind the commas.
  3. 3. Page 4, line 175 "The 175 measurements were repeated in order to obtain reliable data with standard deviation as listed in 176 Tables 2 and 3." Can you specify how many measurements did you do to get the best standard deviation?
  4. 4. Figure 5, the figure is too small and the periodicity is not clear for the wave (the color is too weak). It is better to put the graph like in Figure 6 (with white backgrounds).
  5. 5. Page 7, line 206, "It is widely agreed that 206 material behaves stronger under dynamic loading. Brown concluded through a theoretical 207 formulation that strain rate effect can increase glass strength up to three times". Your result in Table 4 does not reflect this effect.

Author Response

Reviewer 3:

Your article entitled "Measurement of Dynamic Elastic Modulus and 2 Poisson`s Ratio of Chemically Strengthened Glass" is interesting to be published in Materials as the stretching and bending properties of chemically enhanced glass can be used for many applications. However, I found that your manuscript needs to be revised with minor points so that it can be published in this journal.

  1. Why in the beginning you explained three methods (BT, UPET, IET) to determine E, G, and v while the chemically strengthened glass results were only determined by IET. This should be clear in the introduction and Page 7, line 213.

RESPONSE: The corrections have been made in both the introduction and p. 7, line 213 (Sub-section 3.3).

  1. Consistent with decimal points behind the comma. As we see in Table 1 (v and bending stress), it is not clear whether it is 1, 2, or 3 behind the commas.

RESPONSE: Decimal points behind the comma is the result of averaging the values of various mechanical properties listed in Table 1. The corrections has been made in the revised manuscript.

  1. Page 4, line 175 "The 175 measurements were repeated in order to obtain reliable data with standard deviation as listed in 176 Tables 2 and 3." Can you specify how many measurements did you do to get the best standard deviation?

RESPONSE: In addition, it is important to mention here that the measurements were repeated at least 3-4 times in order to obtain reliable data with standard deviation.

  1. Figure 5, the figure is too small and the periodicity is not clear for the wave (the color is too weak). It is better to put the graph like in Figure 6 (with white backgrounds).

RESPONSE: Fig. 5 has been enlarged in the revised manuscript to improve the readability. We would add with white backgrounds, but the images were taken from the software that`s why we are not able to change the background color.

  1. Page 7, line 206, "It is widely agreed that 206 material behaves stronger under dynamic loading. Brown concluded through a theoretical 207 formulation that strain rate effect can increase glass strength up to three times". Your result in Table 4 does not reflect this effect.

RESPONSE: Two sentences have been removed in the revised manuscript.